# Emergent Properties of Foveated Perceptual Systems

## Abstract

The goal of this work is to characterize the representational impact that foveation operations have for machine vision systems, inspired by the foveated human visual system, which has higher acuity at the center of gaze and texture-like encoding in the periphery. To do so, we introduce models consisting of a first-stage *fixed* image transform followed by a second-stage *learnable* convolutional neural network, and we varied the first stage component. The primary model has a foveated-textural input stage, which we compare to a model with foveated-blurred input and a model with spatially-uniform blurred input (both matched for perceptual compression), and a final reference model with minimal input-based compression. We find that: 1) the foveated-texture model shows similar scene classification accuracy as the reference model despite its compressed input, with greater i.i.d. generalization than the other models; 2) the foveated-texture model has greater sensitivity to high-spatial frequency information and greater robustness to occlusion, w.r.t the comparison models; 3) both the foveated systems, show a stronger center image-bias relative to the spatially-uniform systems even with a weight sharing constraint. Critically, these results are preserved over different classical CNN architectures throughout their learning dynamics. Altogether, this suggests that foveation with peripheral texture-based computations yields an efficient, distinct, and robust representational format of scene information, and provides symbiotic computational insight into the representational consequences that texture-based peripheral encoding may have for processing in the human visual system, while also potentially inspiring the next generation of computer vision models via spatially-adaptive computation.

## 1 Introduction

In the human visual system, incoming light is sampled with different resolution across the retina, a stark contrast to machines that perceive images at uniform resolution. One account for the nature of this *foveated* (spatially-varying) array in humans is related purely to sensory efficiency (biophysical constraints) (Land & Nilsson, 2012; Eckstein, 2011), e.g., there is only a finite amount of retinal ganglion cells (RGC) that can relay information from the retina to the Lateral Geniculate Nucleus (LGN) constrained by the thickness of the optic nerve. Thus it is "more efficient" to have a moveable high-acuity fovea, rather than a non-moveable uniform resolution retina when given a limited number of photoreceptors as suggested in Akbas & Eckstein (2017). Machines, however do not have such wiring/resource constraints – and with their already proven success in computer vision (LeCun et al., 2015) – this raises the question if a foveated inductive bias is necessary for vision at all.

However, it is also possible that foveation plays a functional role at the *representational level*, which may confer perceptual advantages – as most computational approaches have mainly focused on saccade planning (Geisler et al., 2006; Mnih et al., 2014; Elsayed et al., 2019; Daucé et al., 2020). This idea has remained elusive in computer vision, but popular in vision science, and has been explored both psychophysically (Loschky et al., 2019) and computationally (Poggio et al., 2014;

| Image | Image Transform: $f_*(\circ)$ | Foveated (Texture-based) Image |
|:---:|:---:|:---:|
| 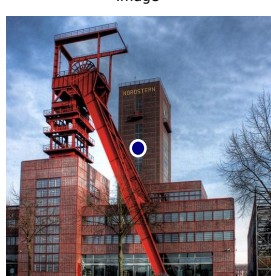 | 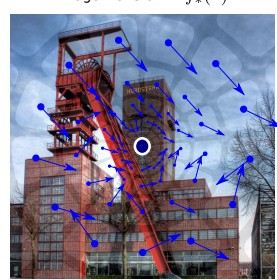 | 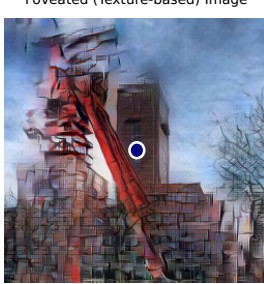 |

Figure 1: A cartoon illustrating how a biologically-inspired foveated image (texture-based) is rendered resembling a human visual *metamer* via the foveated feed-forward style transfer model of Deza et al. (2019). Here, each receptive field is locally perturbed with noise in its latent space in the direction of their equivalent texture representation (blue arrows) resulting in *visual crowding* effects that warp the image locally in the periphery (Balas et al., 2009; Freeman & Simoncelli, 2011; Rosenholtz, 2016). These effects are most noticeable far away from the navy dot which is the simulated center of gaze (foveal region) of an observer under certain viewing conditions.

Cheung et al., 2017; Han et al., 2020). Other works that have suggested representational advantages of foveation include the work of Pramod et al. (2018), where blurring the image in the periphery gave an increase in object recognition performance of computer vision systems by reducing their false positive rate. In Wu et al. (2018)'s GistNet, directly introducing a dual-stream foveal-peripheral pathway in a neural network boosted object detection performance via scene gist and contextual cueing. Relatedly, the most well known example of work that has directly shown the advantage of peripheral vision for scene processing in humans is Wang & Cottrell (2017)'s dual stream CNN that modelled the results of Larson & Loschky (2009) with a log-polar transform and adaptive Gaussian blurring (RGC-convergence). Taken together, these studies present support for the idea that foveation has useful *representational consequences* for perceptual systems. Further, these computational examples have symbiotic implications for understanding biological vision, indicating what the functional advantages of foveation in humans may be, via functional advantages in machine vision systems.

Importantly, none of these studies introduce the notion of *texture representation* in the periphery – a key property of peripheral computation as posed in Rosenholtz (2016). What functional consequences does this well-known texture-based coding in the visual periphery have, if any, on the nature of later stage visual representation? Here we directly examine this question. Specifically, we introduce *perceptual systems*: as two-stage models that have an image transform stage followed by a deep convolutional neural network. The primary model class of interest possesses a first stage image transform that mimics texture-based foveation via *visual crowding* (Levi, 2011; Pelli, 2008; Doerig et al., 2019b,a) in the periphery as shown in Figure 1 (Deza et al., 2019), rather than Gaussian blurring (Wang & Cottrell, 2017; Pramod et al., 2018; Malkin et al., 2020) or compression (Patney et al., 2016; Kaplanyan et al., 2019). These rendered images capture image statistics akin to those preserved in human peripheral vision, and resembling texture computation at the stage of area V2, as argued in Freeman & Simoncelli (2011); Rosenholtz (2016); Wallis et al. (2019).

Our strategy is thus to compare in terms of generalization, robustness and bias these *foveation-texture models* to three other kinds of models. The first comparison model class – *foveation-blur models* – uses the same spatially-varying foveation operations but uses blur rather than texture based input. The second class – *uniform-blur models* – uses a blur operation uniformly over the input, with the level of blur set to match the perceptual compression rates of the foveation-texture nets. Finally, the last comparison model class is the *reference*, which has minimal distortion, and serves as a perceptual upper bound from which to assess the impact of these different first-stage transforms.

Note that our approach is different from the one taken by Wang & Cottrell (2017), who have built foveated models that fit results to human behavioural data like those of Larson & Loschky (2009). Rather, our goal is to explore the emergent properties in CNNs with *texture-based foveation* on scene representation compared to their controls agnostic to any behavioural data or expected outcome. Naturally, the results of our experimental paradigm is symbiotic as it can shed light into both the importance of texture-based peripheral computation in humans, and could also suggest a new inductive bias for advanced machine perception in scenes.

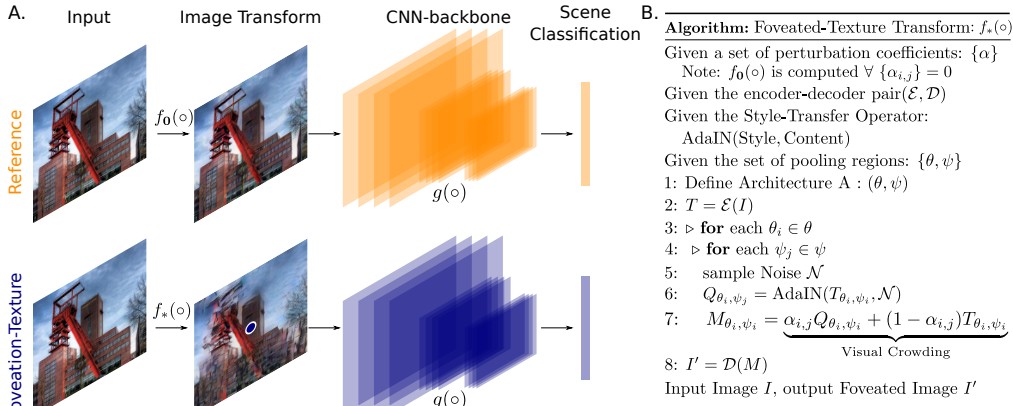

Figure 2: A. Two of the four perceptual systems: Reference (top row) and Foveation-Texture (bottom row), where each system receives an image as an input, applies an image transform ($f(\circ)$), which is then relayed to a CNN architecture ($g(\circ)$) for scene classification. Reference provides an undistorted baseline as a perceptual upper-bound, while Foveation-Texture uses a visual crowding model that distorts the image with spatially-varying texture computation (shown on right) B. The algorithm of how the biologically inspired *Foveation-Texture* transform works which enables effects of *visual crowding* in the periphery (mainly steps 5-7).

## 2 Perceptual Systems

We define perceptual systems as *two-stage* models with an image transform (stage 1, $f(\circ) : \mathbb{R}^D \to \mathbb{R}^D$), that is relayed to a deep convolutional neural network (stage 2, $g(\circ) : \mathbb{R}^D \to \mathbb{R}^d$). Note that the first transform stage is a *fixed* operation over the input image, while the second stage has *learnable* parameters. In general, the perceptual system $S(\circ)$, with retinal image input $I : \mathbb{R}^D$ is defined as:

$$S(I) = g(f(I)) \tag{1}$$

Such two-stage models have been growing in popularity, and the reasons these models are designed to *not* be fully end-to-end differentiable is mainly to *force* one type of computation into the first-stage of a system such that the second-stage $g(\circ)$ must figure out how to capitalize on such forced transformation and thus assess its $f(\circ)$ representational consequences (See Figure 2). For example, Parthasarathy & Simoncelli (2020) successfully imposed V1-like computation in stage 1 to explore the learned role of texture representation in later stages with a self-supervised objective, and Dapello et al. (2020) found that fixing V1-like computation also at stage 1 aided adversarial robustness. At a higher level, our objective is similar where we would like to force a texture-based peripheral coding mechanism (loosely inspired by V2; Ziemba et al., 2016) at the first stage to check if the perceptual system (now foveated) will learn to pick-up on this newly made representation through $g(\circ)$ and make 'good' use of it potentially shedding light on the *functionality* hypothesis for machines and humans.

### 2.1 Stage 1: Image Transform

To model the computations of a texture-based foveated visual system, we employed the model of Deza et al. (2019) (henceforth *Foveated-Texture Transform*). This model is inspired by the metamer synthesis model of Freeman & Simoncelli (2011), where new images are rendered to have locally matching texture statistics (Portilla & Simoncelli, 2000; Balas et al., 2009) in greater size pooling regions of the visual periphery with structural constraints. Analogously, the Deza et al. (2019) Foveation Transform uses a foveated feed-forward style transfer (Huang & Belongie, 2017) network to latently perturb the image in the direction of its locally matched texture (see Figure 1). Altogether, $f : \mathbb{R}^D \to \mathbb{R}^D$ is a convolutional auto-encoder that is non-foveated when the latent space is unperturbed: $f_{\mathbf{0}}(I) = \mathcal{D}(\mathcal{E}(I))$, but foveated ($\circ_\Sigma$) when the latent space is perturbed via localized style transfer: $f_*(I) = \mathcal{D}(\mathcal{E}_\Sigma(I))$, for a given encoder-decoder $(\mathcal{E}, \mathcal{D})$ pair.

Note that with proper calibration, the resulting distorted image can be a visual metamer (for a human), which is a carefully perturbed image perceptually indistinguishable from its reference image (Freeman & Simoncelli, 2011; Rosenholtz et al., 2012; Feather et al., 2019; Vacher et al., 2020). However, importantly in the present work, we exaggerated the strength of these texture-driven distortions

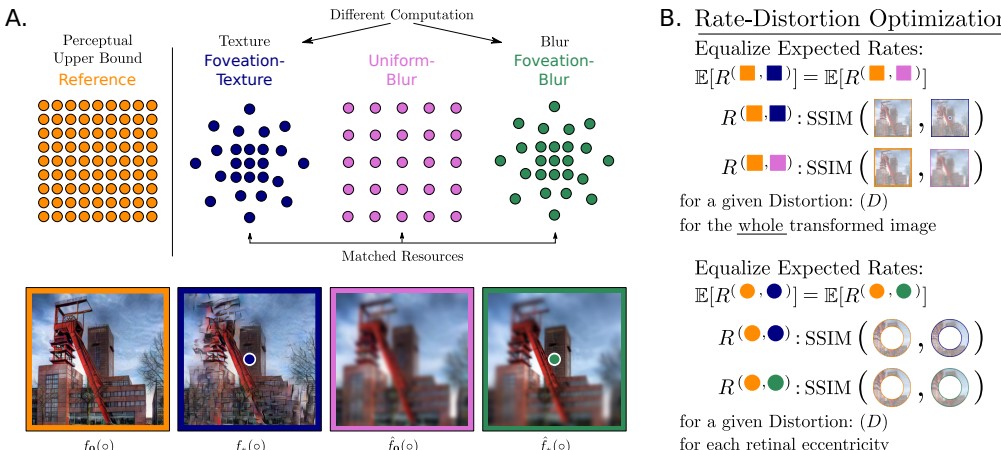

Figure 3: A. Two perceptually matched-resource controls to Foveation-Texture are introduced. Middle-Right, orchid: uniform blurring emulating a matched-resource non-foveated visual system (Uniform-Blur); Far-Right, seagreen: adaptive gaussian blurring (Foveation-Blur) emulating a matched resource blur-based foveated system. B. A Rate-Distortion Optimization procedure is summarized where we find the hyper-parameters of the new matched-resource image transforms $\{(\hat{f}_{\mathbf{0}}(\circ), \hat{f}_*(\circ))\}$ to Foveation-Texture via expected SSIM matching over the validation set.

(beyond the metameric boundary), as our aim here is to understand the implications of this kind of texturized peripheral input on later stage representations (e.g. following a similar approach as Dapello et al. (2020)). By having an extreme manipulation, we reasoned this would accentuate the consequences of these distortions, making them more detectable in our subsequent experiments.

## 2.2 Stage 2: Convolutional Neural Network backbone

The transformed images (stage 1) are passed into a standard convolutional neural network architecture. Here we tested two different base architectures: AlexNet (Krizhevsky et al., 2012), and ResNet18 (He et al., 2016). The goal of running these experiments on two different hierarchically local architectures is to let us examine the consequences across all image transforms (with our main focus towards texture-based foveation) that are robust to these different network architectures. Further, this CNN backbone ($g : \mathbb{R}^D \to \mathbb{R}^d$) should not be viewed in the traditional way of an end-to-end input/output system where the input is the retinal image ($I$), and the output is a one-hot vector encoding a $d$-class-label in $\mathbb{R}^d$. Rather, the CNN ($g$) acts as a loose proxy of higher stages of visual processing (as it receives input from $f$), analogous to the 2-stage model of Lindsey et al. (2019).

## 2.3 Critical Manipulations: Foveated vs Non-Foveated Perceptual Systems

Now, we can define the first two of the four perceptual systems that will perform 20-way scene categorization: *Foveation-Texture*, receives an image input, applies the foveation-texture transform $f_*(\circ)$, and relays it through the CNN $g(\circ)$. Similarly, *Reference* performs a non-foveated transform $f_{\mathbf{0}}(\circ)$, where images are sent through the same convolutional auto-encoder $\mathcal{D}(\mathcal{E}(I))$ of $f_*(\circ)$, but with the parameter that determines the degree of texture style transfer set to 0 – producing an upper-bounded, compressed and non-foveated reference image – then relayed through the CNN $g(\circ)$. Both of these systems are depicted in Figure 2 (A). As the foveation-texture model has less information from the input, relative to the reference networks, we next designed two further comparison models which have a comparable amount of information after the input stage, but with different amounts of blurring in the stage 1 operations. To create matched-resources systems, our broad approach was to use a Rate-Distortion (RD) optimization procedure (Ballé et al., 2016) to match information between the stage 1 operations, given the SSIM (Wang et al., 2004) image quality assessment (IQA) metric.

Specifically, to create matched-resource *Uniform-Blur*, we identified the standard deviation of the Gaussian blurring kernel (the 'distortion' $\mathcal{D}$), such that we could render a perceptually resource-matched Gaussian blurred image – w.r.t Reference – that matches the perceptual transmission 'rate' $\mathcal{R}$ of Foveation-Texture via the SSIM perceptual metric (Wang et al., 2004). This procedure yields a model class with uniform blur across the image, but with matched stage 1 information content as the

Foveation-Texture. And, to create matched-resource *Foveation-Blur*, we carried our this same RD optimization pipeline per eccentricity ring (assuming homogeneity across pooling regions at the same eccentricity), thus finding a set of blurring coefficients that vary as a function of eccentricity. This procedures yielded a different matched-resource model class, this time with spatially-varying blur. Figure 3 (B) summarizes our solution to this problem. Details of the RD Optimization are presented in Appendix A.

Ultimately, it is important to note that the selection of the perceptual metric (SSIM in our case), plays a role in this optimization procedure, and sets the context in which we can call a network "resource-matched". We selected SSIM given its monotonic relationship of distortions to human perceptual judgements, symmetric upper-bounded nature, sensitivity to contrast, local structure and spatial frequency, and popularity in the Image Quality Assessment (IQA) community. However to anticipate any possible discrepancy in the interpretability of our future results, we additionally computed the Mean Square Error (MSE), MS-SSIM, and 11 other IQA metrics as recently explored in Ding et al. (2020) to compare all other image transforms to the Reference on the testing set. Our logic is the following: if the MSE is *greater*($\uparrow$) for Foveation-Texture compared to Foveation-Blur and Uniform-Blur, then the current distortion levels place Foveation-Texture at a resource 'disadvantage' relative to the other transforms, and any interesting results would not only hold but also be *strengthened*. This same logic applies to the other IQA metrics contingent on their direction of *greater* distortion. Indeed, these patterns of results were evident across IQA metrics – except those tolerant to texture such as DISTS (Ding et al., 2020) – as shown in Table 1, and Appendix C.

| (mean±std) | SSIM (*Matched*) | MS-SSIM ($\downarrow$) | MSE ($\uparrow$) | Mutual Information ($\downarrow$) | NLPD ($\uparrow$) | DISTS ($\uparrow$) |
|---|---|---|---|---|---|---|
| Reference | 1.0 | 1.0 | 0.0 | 7.39±0.52 | 0 | 0 |
| Foveation-Texture | $\mathbf{0.58 \pm 0.11}$ | $\mathbf{0.20 \pm 0.03}$ | $\mathbf{976.78 \pm 522.22}$ | $\mathbf{1.40 \pm 0.42}$ | $\mathbf{0.75 \pm 0.16}$ | $0.20 \pm 0.03$ |
| Uniform-Blur | $\mathbf{0.57 \pm 0.15}$ | $0.36 \pm 0.03$ | $458.67 \pm 277.13$ | $1.86 \pm 0.58$ | $0.40 \pm 0.09$ | $\mathbf{0.36 \pm 0.03}$ |
| Foveation-Blur | $\mathbf{0.58 \pm 0.15}$ | $0.36 \pm 0.03$ | $507.35 \pm 302.71$ | $1.84 \pm 0.56$ | $0.45 \pm 0.11$ | $\mathbf{0.35 \pm 0.03}$ |

Table 1: Comparing Image Transforms *wrt* Reference. Arrows indicate direction of *greater* distortion.

## 3 Experiments

Altogether, the 4 previously introduced perceptual systems help us answer three key questions that we should have in mind throughout the rest of the paper: 1) Foveation-Texture vs Reference will tell us how a texture-based foveation mechanism will compare to its perceptual upper-bound – shedding light into arguments about computational efficiency. 2) Foveation-Texture vs Foveation-Blur will tell us if any potentially interesting pattern of results is due to the *type/stage* of foveation. This will help us measure the contributions of the adaptive texture coding vs adaptive gaussian blurring; 3) Foveation-Texture vs Uniform-Blur will tell us how do these perceptual systems (one foveated, and the other one not) behave when allocated with a fixed number of perceptual resources under certain assumptions – potentially shedding light on why biological organisms like humans have foveated texture-based computation in the visual field instead of uniform spatial processing like modern machines.

**Dataset:** All previously introduced models were trained to perform 20-way scene categorization. Scene categories were selected from the Places2 dataset (Zhou et al., 2017), and were re-partitioned into a new 4500 images per category for training, 250 per category for validation, and 250 per category for testing. The categories included were: aquarium, badlands, bedroom, bridge, campus, corridor, forest path, highway, hospital, industrial area, japanese garden, kitchen, mansion, mountain, ocean, office, restaurant, skyscraper, train interior, waterfall. Samples of these scenes coupled with their image transforms can be seen in Figure 4.

**Networks:** Training: Convolutional neural networks of the stage 2 of each perceptual system were trained which resulted in 40 image-transform based networks *per architecture* (AlexNet/ResNet18):

Image Dataset: Mini-Places (20-way Scene Categorization Task)

Reference  Foveation-Texture  Uniform-Blur  Foveation-Blur

(Only 5 scene classes shown in Figure)

Figure 4: Five example images from the 20 scene categories are shown, after being passed through the first stage of each perceptual system.

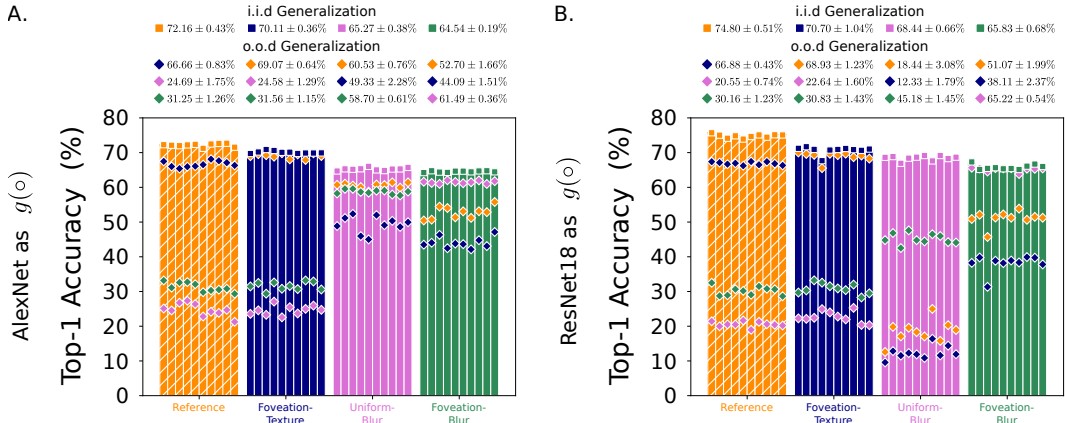

Figure 5: Scene Categorization Accuracy of AlexNet and ResNet18 as $g(\circ)$. We observe the following: Foveation-Texture has greater i.i.d. generalization than other matched-resource systems across both network architectures; Uniform-Blur's o.o.d generalization interacts with the architecture (performing worse for ResNet18, but highest for AlexNet); Foveation-Blur maintains high o.o.d. generalization independent of network architecture. Confusion Matrices can be seen in Appendix I.

10 Foveation-Texture, 10 Reference, 10 Uniform-Blur, 10 Foveation-Blur; totalling 80 trained networks to compute relevant error bars shown in all figures (standard deviations, not standard errors) and to reduce effects of randomness driven by the particular network initialization. All systems were paired such that their stage 2 architectures $g(\circ)$ started with the *same random weight initialization* prior to training. Testing: The networks of each perceptual system were tested on *the same* type of image distribution they were trained on. Learning Dynamics: Available in Appendix H.

### 3.1 Texture-based foveation provides greater *i.i.d.* generalization than Blur-based foveation

How well does the foveation-texture stage classify scene images (i.i.d. generalization) compared to the other matched-resource models that use blurring and the reference? The results can be seen in Figure 5. Each bars' height reflects overall accuracy for each of the 10 neural network backbone runs $(g(\circ))$ per system, with a *square* marker at the top indicating the i.i.d. accuracy. We found that Foveation-Texture had similar i.i.d. performance to the Reference – which is the the undistorted perceptual upper bound, and *greater* performance than both Uniform-Blur and Foveation-Blur. Thus the compression induced by foveated-texture generally maintains scene category information.

We next performed a contrived experiment where we tested how well each perceptual system could classify the stage 1 outputs of the other models. For example, we showed a set of foveated blurred images to a network trained on foveated texture images. This experiment is in essence a test of out-of-distribution *(o.o.d.)* generalization. The results of these tests are also shown in Figure 5. For each model, the classification accuracy for the inputs from the other stage 1 images is indicated by the height of the different colored *diamonds*, where the color corresponds to the stage 1 operation.

This experiment yielded a rather complex set of patterns, that even differed depending on the architecture (AlexNet vs ResNet18 as $g(\circ)$). Generally, the Foveation-Texture model had a similar profile of generalization as the Reference model. However, the networks trained with different types of blur (Uniform-Blur & Foveated-Blur) in some cases showed very high o.o.d. generalization – though once again this is contingent on $g(\circ)$.

Unraveling the underlying causes to understand this last set of results sets the stage for our experiments in the rest of this section. So far it seems like Foveation-Texture has learned to properly capitalize the texture information in the periphery and still out-perform all other matched-resource systems even if heavily penalized under several IQA metrics (Table 1) – highlighting the critical differences in texture vs blur for scene processing. As for the interaction of Uniform-Blur with $g(\circ)$, is is likely that the residual connections are counter-productive to o.o.d. generalization (or it has overfit). Interestingly, humans have a combination of texture and adaptive-gaussian based peripheral computation (Ehinger & Rosenholtz, 2016), so future work should look into the effects of continual learning, joint-training or a combined image transform (Texture + Blur) to merge gains of both i.i.d and o.o.d generalization.

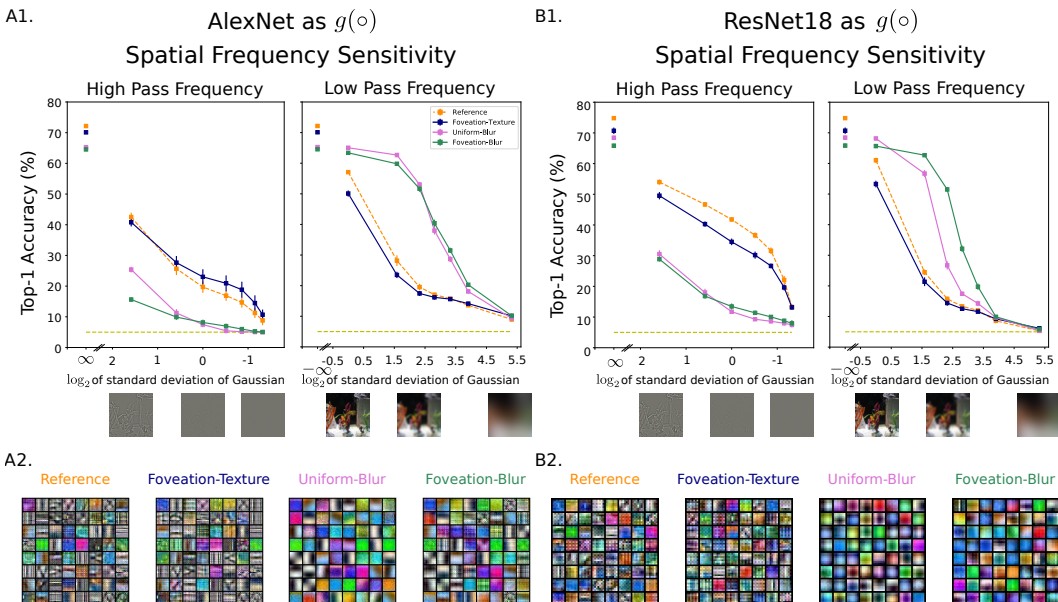

Figure 6: Foveation-Texture has greater sensitivity to high pass spatial frequency filtered stimuli than the Reference (contingent on the architecture for $g(\circ)$ – See A1.,B1.), though both of these systems present notably higher sensitivity to high spatial frequencies than Uniform-Blur and Foveation-Blur. This pattern is reversed for low pass frequency stimuli applied to both color and grayscale filtered images (Appendix K). Visualizations of the first convolutional layer of AlexNet and ResNet18 as $g(\circ)$ (A2.,B2.) shows strong similarities of learned filters despite texture-distortion for Foveation-Texture to Reference preserving high spatial frequency Gabors; Uniform-Blur shows a strong predominance of low spatial frequency Gabors for AlexNet and low spatial frequency center-surround filters for ResNet18, and Foveation-Blur a mixture of high-low spatial frequency tuned filters.

## 3.2 Texture-based foveated systems preserve greater high-spatial frequency sensitivity

We next examined whether the learned feature representations of these models are more reliant on low or high pass spatial frequency information. To do so, we filtered the testing image set at multiple levels to create both high pass and low pass frequency stimuli and assessed scene-classification performance over these images for all models, as shown in Figure 6. Low pass frequency stimuli were rendered by convolving a Gaussian filter of standard deviation $\sigma = [0, 1, 3, 5, 7, 10, 15, 40]$ pixels on the foveation transform $(f_\mathbf{0}, \hat{f}_\mathbf{0}, f_*, \hat{f}_*)$ outputs. Similarly, the high pass stimuli was computed by subtracting the reference image from its low pass filtered version with $\sigma = [\infty, 3, 1.5, 1, 0.7, 0.55, 0.45, 0.4]$ pixels and adding a residual. These are the same values used in the experiments of Geirhos et al. (2019).

We found that Foveation-Texture and Reference trained networks were more sensitive to High Pass Frequency information, while Foveation-Blur and Uniform-Blur were selective to Low Pass Frequency stimuli. Although one may naively assume that this is an expected result – as both Foveation-Blur and Uniform-Blur networks are exposed to a blurring procedure – it is important to note that: 1) the foveal resolution has been *preserved* between Foveation-Texture and Foveation-Blur (See Fig. 4), thus high spatial frequency sensitivity could have still predominated in Foveation-Blur but it did not (though see Fig. 6 A2/B2 where these high pass Gabors are still learned, implying that higher layers in $g(\circ)$ overshadow their computation); and 2) Foveation-Texture could have also learned to develop low spatial frequency sensitivity given the crowding/texture-like peripheral distortion, but this was not the case (likely due to the weight sharing constraint embedded in the CNN architecture Elsayed et al., 2020). Finally, the robustness to low-pass filtering of Foveation-Blur suggests that foveation via adaptive gaussian blurring may implicitly contribute to scale-invariance as also shown in Poggio et al. (2014); Cheung et al. (2017); Han et al. (2020).

## 3.3 Texture-based foveation develops greater robustness to occlusion

We next examined how all perceptual systems could classify scene information under conditions of visual field loss, either from left to right (left2right), top to bottom (top2bottom), center part of

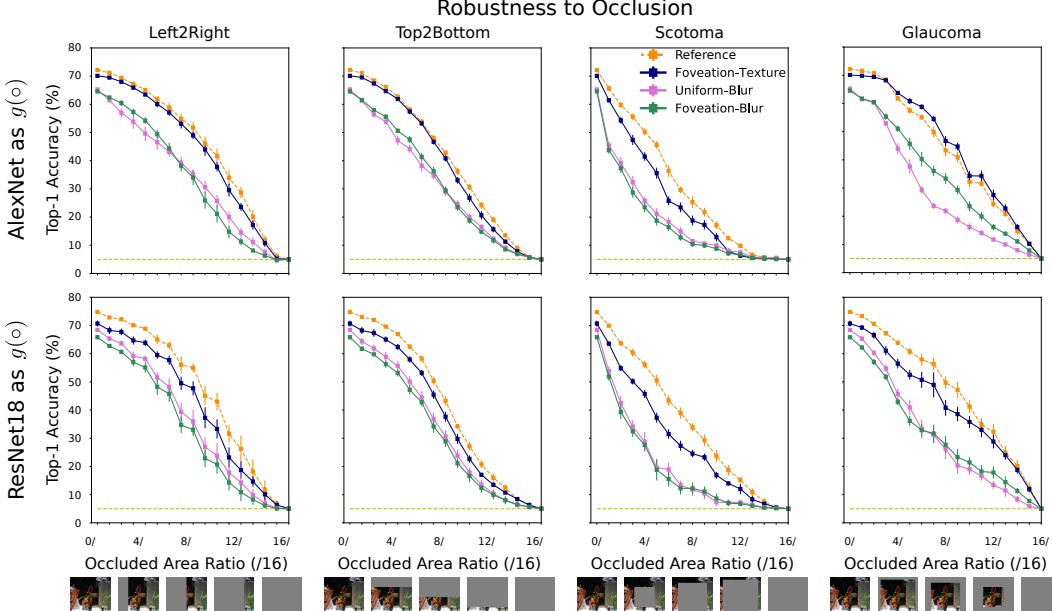

Figure 7: Foveation-Texture has greater robustness than both Foveation-Blur and Uniform-Blur while roughly preserving a performance similarity to Reference (the upper bound) beyond the *i.i.d.* regime. The asymmetry in performance of the Scotoma vs Glaucoma conditions for foveated models also suggests they have learned to weigh spatial information differently in the fovea vs the periphery despite a weight sharing constraint imposed through $g(\circ)$.

the image (scotoma), or the periphery (glaucoma). This manipulation lets us examine the degree to which learned representations relying on different parts of the image to classify scene categories. Critically, here we apply the occlusion *after* the stage 1 operation. The results are shown in Figure 7.

Overall we found that, across all types of occlusion the Foveation-Texture modules have greater robustness to occlusion than both the Foveation-Blur and Uniform-Blur models. Further, the Foveation-Texture models have nearly equivalent performance to the Reference. In contrast, both models with blurring, whether uniformly or in a spatially-varying way, were far worse at classifying scenes under conditions of visual field loss. These results highlight that the texture-based information content captured by the foveation-texture nets preserves scene category content in dramatically different way than simple lower-resolution sampling – perhaps using the texture-bias (Geirhos et al., 2019) in their favor; as humans too use texture as their classification strategy for scenes (Renninger & Malik, 2004).

In addition, the Foveation-Texture model is not overfitting. As recent work has suggested an Accuracy vs Robustness trade-off where networks trained to outperform under the *i.i.d.* generalization condition will do worse under other perceptual tasks – mainly adversarial (Zhang et al., 2019) – we did not observe such trade-off and a greater accuracy did not imply lower robustness to occlusion.

### 3.4 Foveated systems learn a stronger center image bias than non-foveated systems

It is possible that foveated systems weight visual information strongly in the foveal region than the peripheral region as hinted by our occlusion results (the different rate of decay for the accuracy curves in the Scotoma and Glaucoma conditions). To resolve this question, we conducted an experiment where we created a windowed cue-conflict stimuli where we re-rendered our set of testing images with one image category in the fovea, and another one in the periphery (all aligned with a different class systematically; *ex:* aquarium with badlands). We also had an additional condition where the conflicting cue was now square-like and uniformly and randomly paired with a conflicting scene class and more finely sampled. We then systematically varied the fovea-periphery visual area ratios & re-examined classification accuracy for both the foveal and peripheral scenes (Figure 8).

We found that the Foveation-Texture and Foveation-Blur transform imposed the networks $g(\circ)$ to learn to weigh information in the center of the image stronger than Reference & Uniform-Blur for scene categorization. A qualitative way of seeing this foveal-bias is by checking the foveal/peripheral

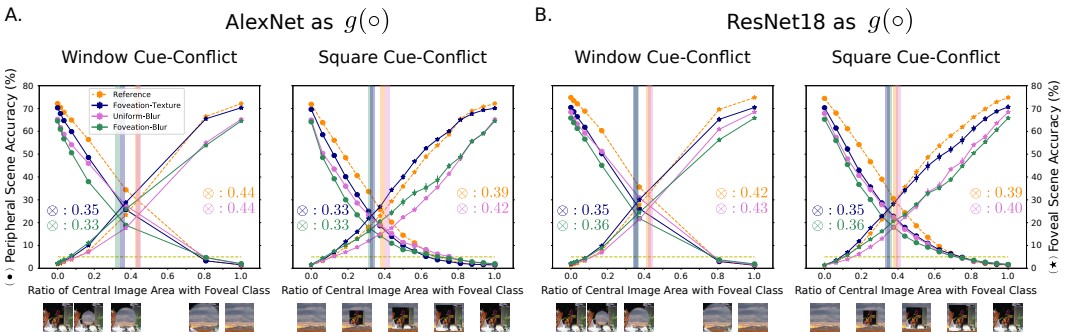

Figure 8: Foveated Perceptual Systems – independent of the computation type (Foveation-Texture, Foveation-Blur) – show stronger biases to classify hybrid scenes with the foveal region; a result also observed in humans (Larson & Loschky, 2009).

ratio where these two accuracy lines cross. The more leftward the cross-over point (⊗), the higher the foveal bias (highlighted through the vertical bars). This result was unexpected as we initially predicted that $g(\circ)$ would weigh the peripheral information stronger as it has been implicitly regularized through a distortion. However this was not the case and our findings are similar to Wang & Cottrell (2017) who showed this foveal bias on a foveated system with adaptive blur with a dual-stream neural network. Thus, these results indicate that the *spatially varying computation from center to periphery* is mainly responsible for the development of a center image bias *even with a weight sharing constraint*. Furthermore, it is possible that one of the functions of any spatially-varying coding mechanisms in the visual field is to *enforce* the perceptual system to *attend* on the foveal region – avoiding the shortcut of learning to attend the entire visual field if unnecessary (Geirhos et al., 2020).

## 4 Discussion

The present work was designed to probe the impact of foveated texture-based input representations in machine vision systems. To do this we specifically compared the learned perceptual signatures in the second-stage of visual processing across a set of of networks trained on other image transforms. We found that when comparing Foveation-Texture to their matched-resource models that differed in computation: Foveation-Blur (foveated w/ adaptive gaussian blur) and Uniform-Blur (non-foveated w/ uniform blur) – that peripheral texture encoding did lead to specific representational signatures, particularly greater i.i.d generalization, preservation of high-spatial frequency sensitivity, and robustness to occlusion – even as high as its perceptual upper bound (Reference). We also found that foveation (in general) seems to induce a *focusing mechanism*, servicing the foveal/central region – whereas neither a perceptually upper-bounded system (Reference) or a non-foveated compressed system (Uniform-Blur) did *not* develop as strongly.

The particular consequences of our foveation stage raises interesting future directions about what computational advantages could arise when trained on object categorization (Pramod et al., 2018) coupled with eye-movements (Akbas & Eckstein, 2017; Deza et al., 2017), as objects are typically centered in view and have different hierarchical/compositional priors than scenes (Zhou et al. (2014); Deza et al. (2020)) in addition to different processing mechanisms (Renninger & Malik (2004); Ehinger & Rosenholtz (2016)). We are currently exploring the impact of these *foveated texture-based* representational signatures on shape vs texture bias for object recognition similar to Geirhos et al. (2019) and Hermann et al. (2020), and assessing their interaction with scene representation.

Further, a future direction is investigating the effects of texture-based foveation to *adversarial robustness*. Motivated by the recent work of Dapello et al. (2020) which has shown promise of adversarial robustness via enforcing stochasticity and V1-like computation by obeying the Nyquist sampling frequency of these filters w.r.t the image (Serre et al., 2007) in addition to a natural gamut of orientations and frequencies as studied in De Valois et al. (1982), it raises the question of how much we can further push for robustness in hybrid perceptual systems like these, drawing on even *more* biological mechanisms. Works such as Luo et al. (2015) and recently Reddy et al. (2020); Kiritani & Ono (2020) have already taken steps in this direction by coupling fixations with a spatially-varying retina. However, the representational impact of texture-based foveation on adversarial robustness, and its symbiotic implication for human vision still remains an open question.

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
