# OpenReview forum: "Emergent Properties of Foveated Perceptual Systems"
_NeurIPS.cc/2021/Conference — NeurIPS 2021 Submitted_

### Official Review · Reviewer_89Wr · 2021-07-14

**Rating:** 5
**Confidence:** 4

**Summary:**

In this manuscript the authors test how DNNs trained on classifying scene categories react to texture based foveation, which they compare to undisturbed images, a foveated blur and a uniform blur matched in their SSIM to the original image. In overall accuracy the texture model performs close to the performance of the undistorted images and clearly better than the other two distortions. Going beyond accuracy the authors then test the effects of the input distortions on processing filtered images, occlusion, and cue conflicts between center and periphery.


**Limitations And Societal Impact:**

The authors do not comment on their limitations and societal impact. Besides the sampling biases of the used databases I don’t think there are important societal impacts associated with the content of this paper though.

**Main Review:**

Overall, I am torn on this one. On the one hand, the authors did a lot of work and I don’t have many technical issues with their work. On the other hand, I do not see at all what the relevance of the performed experiments is or what could be learned from them. The texture based foveation the authors apply appears quite different from the way we think the human visual system implements this, which limits any interpretation towards human comparisons and the observed effects of the foveation seem rather trivial. This leaves me with the question: yes, but why?

To elaborate on the human comparison problem: This kind of peripheral texturization of images is a method for testing human peripheral perception, which is thought not to differentiate these texturized images from each other due to computing summary statistics at midlevel processing steps. Using such distorted images as input to a non-foveated processing system really does not capture the human foveated processing well in my opinion. Also this implementation provides none of the advantages of foveated processing like reduced computational power requirements for the periphery.

And on the observations not being particularly interesting:

Figure 6: Networks trained with blurred inputs react more to lower frequencies. Although, the authors dismiss this as a naive idea (in 3.2), I still find this unsurprising. Even if some higher frequency information in available for the foveated blur, it certainly sees much more low frequency content.

Figure 7: In principle, better performance for occlusions is an independent observation, but after observing a generally better performance of the Foveation-texture model this mostly looks like this advantage is simply preserved across occlusions.

Figure 8: Models trained with more information in the center weigh central information slightly stronger. That is definitely expected and if anything the effect is surprisingly small due to the non-foveated processing in the models.


**Time Spent Reviewing:**

5

---

> ### Author Response · Authors · 2021-08-10
> **Response to Reviewer 89Wr [1/1]**
>
>
> Thank you for your time and energy with regards to submitting your review! We hope that our rebuttal successfully addresses several of the lingering questions you may have, and can thus encourage you to raise your score. Please let us know if we may have missed something!
>
> * " [...] The texture based foveation the authors apply appears quite different from the way we think the human visual system implements this '"
>
> We disagree, please see Balas, Nakano \& Rosenholtz (2009); Freeman \& Simoncelli, 2011 (Metamers); Rosenholtz 2011 (Mongrels); Wallis, Bethge \& Wichmann, 2016 (Testing Peripheral processing); and recently Wallis et al. 2019 (scene content is more important than Bouma's law for Scene metamers).
>
> * "This kind of peripheral texturization of images is a method for testing human peripheral perception, which is thought not to differentiate these texturized images from each other due to computing summary statistics at midlevel processing steps. **Using such distorted images as input to a non-foveated processing system really does not capture the human foveated processing well in my opinion**.'"
>
> Indeed, our goal was **not to create a model of human vision**. The goal of this paper is to test if a foveated texture-based inductive bias (inspired by human peripheral vision) on a machine could give rise to different perceptual signatures than a set of 2 non-foveated inductive biases with different rates of compression, or a matched-resource adaptive-gaussian based inductive bias. Our goal was not to beat a state-of-the-art model as done in classical computer vision papers -- and neither to present a biologically plausible model of the human visual system to fit our model to behaviour as done in vision science; rather our goal is to show precise and empirical understanding of what are the potential consequences of different types of human-inspired visual processing (foveation) in machines -- and we think NeurIPS is the most appropriate audience for such type of works.
>
> * "[... Trivial] Even if some higher frequency information in available for the foveated blur, it certainly sees much more low frequency content."
>
> We disagree, this result is non-trivial as it could have been that due to the weight sharing constraint **and the learned central image bias** (which is high resolution; see Figure 8), that the network would behave more than Foveation-Texture in terms of spatial frequency sensitivity (despite its heavy distortions). This was in fact our prediction, and we were suprised how much more similar it was to just uniform-blur that does not have multi-resolution inputs. Recall: None of the networks are trained with any data-augmentation such as random cropping or re-scaling that could implicitly induce low-spatial resolution bias.
>
> * "[... Trivial] but after observing a generally better performance of the Foveation-texture model this mostly looks like this advantage is simply preserved across occlusions. '"
>
> We disagree that this is a trivial result: it could have been the case that the Foveation-Texture (or any other model) could have over-fitted at training time. Some works have shown an Accuracy vs Robustness trade-off where models with **higher** accuracy will have **lower** robustness to distortions [Zhang et al., ICML 2019]. We did not see such effect in our experiments, and this pattern was preserved over the learning dynamics (see Supplementary material).
>
> * "[... Trivial] Models trained with more information in the center weigh central information slightly stronger. That is definitely expected and if anything the effect is surprisingly small due to the non-foveated processing in the models. "
>
> This is an interesting observation that perhaps falls in the "hindsight is 20/20" area as we disagree that it is initially obvious. Studies in scene perception by Larson \& Loschky 2009 have shown the importance of the peripheral region when performing scene recognition as it provides a big contextual cue when performing the task (also recall Oliva \& Torralba's scene gist). It could been the case that manipulating the peripheral region regularized the scene and forced networks to pick-up on such cues to provided more context -- but this did not happen. Indeed, a previous statement by the reviewer was ([...] "Foveation-Blur has more low pass area, so it is obvious it will have a low-pass bias"); under the same line of thought one could claim that a peripheral bias should have emerged rather than a center bias, but this was not the case.
>
> Finally, recall that there is a weight-sharing constraint for these networks, and it is not trivial or obvious that the higher non-convolutional layers would implicitly re-organized the systems to ``attend'' to the center even with different sets of learned filters and features that are shared through the visual hierarchy.
>
> * "Also this implementation provides none of the advantages of foveated processing like reduced computational power requirements for the periphery."
>
> Showing that foveation provides computational efficiency in this paper is not our goal as this has already been shown when coupled with Eye-Movements for object detection [Akbas \& Eckstein, 2017] and without eye-movements for scene perception [Lester \& Loscky, 2009]. Complimentarily, we wanted to show that there is perhaps a **representational goal** in **texture-based** foveation that mimic the type of peripheral processing performed in humans. If a family of scene statistics align with texture-like processing, then it is (likely) no accident that the human visual system was designed with a different type of processing mechanism in the visual periphery rather than the fovea. **To our knowledge no one has proved this with an adaptive texture-based computational model of foveation** -- which is a side-bonus of our paper as our focus is mainly centered on the impact (learned representational signatures) of foveation on machine perception.
>
> We wanted to thank you once again for your review, and please let us know if there is any other points to clarify or potential follow-up experiments that may persuade you to raise your score.

---

> > ### Comment · Reviewer_89Wr · 2021-08-15
> > **acknowledging response**
> >
> > I just wanted to shortly write to acknowledge the authors response although it didn't change my rating of this paper.
> >
> > Just to clarify things: My main concern with this manuscript remains and is reflected in the other reviews as well: Instead of implementing the peripheral pooling which is thought to cause textures to seem similar to normal images in the periphery, the authors choose to produce images with texturized peripheries to a model which is uniform over space. Even the positive reviewer qxyi calls this a peculiar choice.
> >
> > I don't think the match between a model receiving peripherally texturized images as input and a model with peripheral pooling is good enough to spark interest in the behavior of the former. Unfortunately, this is not changed by further investigating this former model.

---

> > > ### Author Response · Authors · 2021-08-15
> > > **Following up with Reviewer 89Wr, Claryfing Peripheral Pooling [1/1]**
> > >
> > > Thank you for your reply Reviewer 89Wr! We are following up on these very valid concerns in reverse order:
> > >
> > > * “I don't think the match between a model receiving peripherally texturized images as input and a model with peripheral pooling is good enough to spark interest in the behavior of the former. Unfortunately, this is not changed by further investigating this former model.”
> > >
> > > **We am not sure what is meant by the reviewers' suggestion to implement “peripheral pooling”?  [And we are also a bit confused]** This was done in Deza & Eckstein (NeurIPS 2016), but this type of feature pooling does not give rise to texture-based representations. Also we *do* want $g(\circ)$ to have uniform convolutions across all systems as it is supposed to act as a higher stage of visual processing (See Section 2.2 Lines 113-122) and provides a stronger control wrt to the non-foveated systems. In addition, one would need to do a distortion in the latent space in the direction of the texture via a style-transferred noise image as in Deza, Jonnalagadda & Eckstein; (ICLR 2019) to property encode for texture. It is also the decoder from that transform that learns to interpret that distortion as a texture which is why we preferred to do this in image space.
> > >
> > > * " Just to clarify things: My main concern with this manuscript remains and is reflected in the other reviews as well: Instead of implementing the peripheral pooling which is thought to cause textures to seem similar to normal images in the periphery, the authors choose to produce images with texturized peripheries to a model which is uniform over space. Even the positive reviewer qxyi calls this a peculiar choice. "
> > >
> > > This is a valid concern --and we are doing this for a follow-up paper where we are exploring adversarial robustness in a fully-differentiable foveated texture-based transform. We’d argue that for our current paper our implementation is ideal given that the interpretation is that the model is really $S(I)$ that is composed of a fixed 1st-stage transform $f(\circ)$ and a learnable 2nd-stage $g(\circ)$ s.t. $S = g(f(I))$. Having $f$ as an image to image-to-image transform allows for a guarantee of foveated texture-based processing and should show more convincing results. For example the well known Texture-Bias paper of Geirhos et al. ICLR 2019 imposes texture invariance by applying altering the texture with style transfer: $f(\circ)$ (also Adaptive Instance Normalization) on the image dataset and then training a CNN $g(\circ)$-- they could have also encoded texture invariance in the feature space of the deep neural network, but the fact that they performed it on the image sheds no doubt that: 1) the type of texture-based transform was property implemented, and 2) the texture-bias exists/emerges on a non-modified CNN. There are 2 other reasons for us as well:
> > >
> > > 1) We can define $f(\circ)$ as *any* type of image-to-image transform which is ideal for our **control conditions** such as uniform blur and adaptive blur. Doing this in the feature space would have been non-trivial as we would potentially run into issues with regards to matching dimensionality and confounds due to different numbers of parameters.
> > > 2) Given point (1), *performing a Rate-Distortion Optimization on this new image space (output of $f$) is computationally tractable* and quantifiable given IQA metrics. Otherwise, it would have been non-trivial  -- and perhaps non-tractable -- to do a feature-space optimization between all 4 models (One could perhaps use an L2 based loss, *but how could we check if they are correctly matched if we can not visualize the images?*)
> > >
> > > **We hope these follow-up comments clarify your doubts, but please let us know if something still remains unclear! Perhaps there is a confusion on the goal (rather than the method) of our paper?**

---

> > > ### Comment · Area_Chair_eer9 · 2021-08-25
> > > **Authors' response would be helpful**
> > >
> > > To the authors -- the comment from this reviewer is a potential death sentence for this paper... could you please try to respond? Thanks

---

> > > > ### Author Response · Authors · 2021-08-25
> > > > **Reponse has been issues below**
> > > >
> > > > Dear AC,
> > > >
> > > > Thank you for your response, we did respond to the reviewer's follow-up comments on August 15th (10 days ago) as seen in our post-rebuttal comment here (please confirm if it is visible!):
> > > > https://openreview.net/forum?id=cdlzxKUBr-A&noteId=of1U2Bv-FII
> > > >
> > > > The Authors

---

### Official Review · Reviewer_ThVH · 2021-07-15

**Rating:** 3
**Confidence:** 3

**Summary:**

The paper aims to explore the effects on representation that emerge when a CNN is equipped with a foveated visual system. The authors build a model whereby images are artificially rendered into a form that captures foveation which are then fed into standard CNN architectures. Two different types of foveation are considered: one where the periphery is blurred (greater blurring as a function of eccentricity), and another where texture statistics are preserved over local regions which increase in size as a function of eccentricity.  These are compared against each other and two baselines:

- Reference: images are transformed by pushing them through a convolutional AE
- Foveation-texture: images are transformed by pushing them through a convolutional AE with a foveated texture style transfer
- Uniform-blur: each image is blurred uniformly, with the amount of blurring set such that the SSIM between the blurred image the its corresponding reference was the same as the SSIM between the corresponding foveation-texture image and reference.
- Foveation-blur: as above, but with blur additionally increasing with eccentricity

Clearly the reference images contain more information than the foveation-texture images; the SSIM matching on the is done to resource match the blurred images for comparison to foveation-texture, and the reference serves as an upper bound. Experiments are performed on a scene classification task.

**Limitations And Societal Impact:**

Limitations are not adequately addressed as I've described above. It's probably fair to say that there are no major potential negative societal impacts, although I'd note that the energy cost in training all of these models must have been quite large (in terms of CO2 equivalent).


**Main Review:**

Originality
-----------

The paper is original in the sense that it's trying to give a new understanding of the potential benefits of a foveated visual system. I believe it is adequately cited and is complementary in its aims to prior work.

Quality
-------

Clearly a lot of work has gone into this paper (just from the the shear number of experiments performed for example), however I have a number of questions/issues which I'll describe below. It is of course possible that I've misinterpreted some parts of the paper, and if so, I look forward to a discussion with the authors during that phase of the review process.

__The model:__

Two-stage models like the one proposed (and e.g. Lindsay (2019) as mentioned, but also recent work looking at pre-filtering the network input with Gabor filters or similar) have become popular ways of exploring how different inductive biases affect representations in artificial visual systems. To the best of my knowledge however, all the prior work has involved using an input that is matched with the inductive biases built into the network - that is the spatial resolution of the input is constant and thus matched to the convolutions that follow. In this work, the foveated networks effectively vary spatial resolution across spatial position, so it is not in anyway clear that applying fixed convolutions (with respect to the space between taps) across this input in anyway makes sense. Have the authors considered this? Why wasn't an alternative (perhaps multiple streams of fixed spatial resolution in the 1st stage? or maybe not using shared weights? or...).

With respect to the resource matching, it isn't clear to me why matching using a perceptual metric is the correct approach. Surely the aim should be to match the amount of information that is transmitted to the second stage? in which case why wouldn't you e.g. match based on entropy?


__The results:__

The authors have chosen to do their experiments with a variant of the Places2 dataset for a scene categorization task, but there isn't really any discussion of why this was chosen in the paper, or of any of the potential pitfalls of using this dataset in the evaluation that is performed. By its very nature a scene categorization task requires information from across the entire image, at least in expectation over the entire dataset; for example, many (the majority?) of the images have very human biases which affect their overall composition - e.g.:

- photos with a person in the foreground, often near the center of the image, where the presence of a person is unrelated to the scene (e.g. ocean)
- images where if you relied just on the center you would be unlikely to predict the 'correct' scene (the badlands image in the paper is a good representative example)

These factors need to be considered when analysing results of experiments, because they will inherently cause a bias. In this case before doing any experiments, one might reasonably expect that if the periphery of the image is corrupted in some way performance would be lower. Further, because early work on scene classification (e.g. using things like bags of visual words with quantised dense SIFT descriptors, which were basically capturing local texture) actually worked rather well on these tasks, one would inherently expect that texture is going to be an important factor. With this in mind, I question if many of the results in the paper actually tell you more about the nature of the data rather than about foveation?

I've written comments/questions on each experimental section below:

_"Texture-based foveation provides greater i.i.d generalisation than Blur-based foveation"_

- The above claim could possibly be inferred from the results, but it could just be that you've shown that the task you've chosen is more easily solved with high-frequency/texture information as I suggested previously? Do the differences between uniform and foviated -blur just indicate that for this task there is _relatively more_ information to solve the task in the periphery (which shouldn't really be surprising given the nature of the dataset)?
- The claim that foveation-texture has "similar i.i.d performance" to reference seems like something of an overstatement; it's closer than blur, but still seems worse (particularly for the resnet) unless I'm missing something?
- Following on, isn't it obvious that texture is likely to be more informative given what is already known about this type of dataset?
- The reference images applied to foveated-texture showing good generalisation seems intuitive, however the uniform blur differences across networks are odd. When I looked at this I the question that arose is "why is alexnet so good (particularly w.r.t reference - you'd expect foviated-blur to be better)?"; the authors clearly thought that problem was instead with ResNet (which I'd argue has more intuitive performance with respect to the ordering of each input type) - I wonder if the authors could elaborate on this?

_"Texture-based foveated systems preserve greater high-spatial frequency sensitivity"_

- I don't understand the argument in note 1: yes, the foveal bit is the same, but the texture model clearly has more high frequencies, which for this task are more likely to be important. Point 2 again just suggests that higher frequencies are more important for this task.

_"Texture-based foveation develops greater robustness to occlusion"_

- So the takeaway is again that the solution to the task is dominated by higher frequency information, and clearly there is relevant high-frequency information in the periphery that helps significantly? Again, given the data is this really surprising?

_"Foveated systems learn a stronger center image bias than non-foveated systems"_

- I'm completely failing to understand what is surprising about this (or why the authors hypothesised it would be different); surely any learning system given a mixture of true and slightly corrupted data is going to have a preference for weighting the uncorrupted data higher?


Clarity
-------

From a technical perspective I believe enough information is given to be able to reproduce the experiments if one were minded. I did however struggle with understanding the motivation of the work (largely with respect to the model choices as outlined above). I also felt that the introduction was not sufficiently self-contained - for example the reader needs to be fully aware of the contents Rosenholtz (2016) to really understand some of the specific motivation for the _foveation-texture_ model (I had to read it; perhaps it is "well known" in the core vision sciences community, but this is a paper submitted to NeurIPS where the audience is much broader (particularly with respect to people in the computer vision community, as well as those more generally interested in understanding how different biases affect representation). I would suggest that this bit of the paper needs rewriting to make it much more self contained.


Significance
------------

Commenting on the significance is tough: having read the paper, do I know things that I didn't before? yes. Are they useful? that's completely unclear to me. As I've outlined above it's not clear to me that the model itself makes sense, or that the results actually tell us anything other than that the data used to train/evaluate the models had an inherent 'human' bias which can be exploited by foveation.

**Time Spent Reviewing:**

7

---

> ### Author Response · Authors · 2021-08-10
> **Response to Reviewer ThVH [1/1]**
>
>
> Thank you for your time with regards to sending your review. Hopefully we have addressed all your concerns in the following response -- specially with regards to the goal, the methods, and clarity --, and please let us know if something is still unclear, given your high enthusiasm about the interdisciplinary nature of our paper and its potential value for NeurIPS.
>
>
> * Model: " [...] so it is not in anyway clear that applying fixed convolutions (with respect to the space between taps) across this input in anyway makes sense. Have the authors considered this? Why wasn't an alternative (perhaps multiple streams of fixed spatial resolution in the 1st stage? or maybe not using shared weights? or...)."
>
> This was considered, but we decided not to make it an alternative as we needed a guarantee of a rendered texture in the image that has been psychophysically tested (Deza et al. 2019).  However we do agree that re-adapting the interior wirings of the networks with localized Gramian-like structure is a possibility a la Gatys et al. (this is current work as we are designing a fully differentiable foveated model to test such systems against adversarial attacks where end-to-end differentiability is necessary) -- however this is not the focus of the current paper: **Our goal is to test and understand the effects of different foveated and non-foveated image transforms on machine vision systems** -- with the hope of symbiotically finding evidence for a representational goal for texture-based computation in humans simulated through an image transform.  In addition not adding more biologically plausible constraint in our model also provides stronger controls when comparing across perceptual systems as the architecture is held constant and the input is systematically varied.
>
> * RD-Optimization: " [...] With respect to the resource matching, it isn't clear to me why matching using a perceptual metric is the correct approach. Surely the aim should be to match the amount of information that is transmitted to the second stage? in which case why wouldn't you e.g. match based on entropy?"
>
> We did in fact compute average Mutual Information from the luminance values of each transform and found the Foveation-Texture transform to be at a disadvantage compared to the other 2 models (Table 1), and in general we found that the perturbations done on Foveation-Texture greater than the Uniform-Blur and Foveation-Blur across nearly all IQA metrics strengthening our results. The problem with metrics like Mutual Information and Entropy is that a fully scrambled image and an original yield the same entropy-based score making it inappropriate for our experiments (!)
>
> * "These factors [center photographical bias and choice of scene dataset] need to be considered when analysing results of experiments, because they will inherently cause a bias. "
>
> This is a highly relevant observation and we absolutely agree! This point has been addressed throughout the paper via the **Reference-Net** dashed line baseline that will absorb any existing dataset biases.
>
> * "By its very nature a scene categorization task requires information from across the entire image.''
>
> We agree; and this is the main reason why we chose scenes rather than objects for our experiments. Perhaps we have missed the point addressed by the reviewer? We'd like to make it clear that our choice of dataset (scenes) **does** in fact play a critical role in our results as it is only appropriate to study emergent representations of learnable models with **foveation** when working with scenes.
>
> * "Motivation for Mini-Places dataset [... instead of Places or ImageNet]''.
>
> Learning Theory Motivation: The motivation for using a modified Places dataset is to have a strict partition of **equal** number of training images per class -- potentially avoiding a bias towards a specific class prediction as usually happens when training networks on unbalanced datasets like ImageNet (120 dogs classes from 1000 objects) and/or Places (imbalanced number of samples contingent on scene class). This was initially added in our manuscript but removed due to space constraints. We will add this in the extended version of our manuscript with the additional page.
>
> Vision Science Motivation: Scenes were used instead of objects given that scene recognition in humans involve using a wide field of view even if the camera is not centered on an object [Loshcky, et al. 2017, Larson \& Loschky, 2009; Oliva and Torralba, 2001; Ehinger \& Rosenholtz ,2016]; rather than object recognition where humans place objects in their small central field of view **(the fovea)** given object size [DiCarlo et al. 2013; Logothetis and Paul (1995)]. In that regard, "foveating" an image from a cropped object (for example if using ImageNet), would not be biologically plausible [and would in fact give an un-realistic setup for testing mechanisms of peripheral vision] given that the foveated transform is performing strong distortions in the outer visual field -- that for object centered images like ImageNet -- heavily distort the object as it occupies 80\% of the image. In general, core **object recognition** under normal viewing takes place in the foveal/central region, while **scene recognition** in normal viewing takes up the entire field of view. This is the motivation for using a scene-based dataset.
>
> * "The claim that foveation-texture has "similar i.i.d performance" to reference seems like something of an overstatement; it's closer than blur, but still seems worse (particularly for the resnet) unless I'm missing something?''
>
> We do not think this is an over-statement as it is closer to Reference-Net **compared to all other models** in terms of i.i.d performance. We will add this clarification.
>
> * "Following on, isn't it obvious that texture is likely to be more informative given what is already known about this type of dataset?''
>
> We were actually quite surprised, despite texture computation being related to scene recognition; the fact that the texture-based model loses **structure** in the periphery and still outperforms the Uniform-Blur or Foveated-Blur neural networks is quite surprising. Furthermore, even though we did our RD-Optimization with SSIM, nearly all other IQA metrics (See Supplement Table 2) showed that the texture-based distortions were **greater** for our texture based model in comparison to the other 2.
>
> Perhaps a better way of articulating our views is that we knew the foveated texture model would classify scenes correctly, but not this good(!) We think this is a "hindsight is 20/20'' type observation made both by us and the reviewers.
>
> * "why is alexnet so good (particularly w.r.t reference - you'd expect foviated-blur to be better)?"; the authors clearly thought that problem was instead with ResNet ''
>
> It is not that we think ResNet is problematic, it is just that the architecture induces a different type of representational space when solving the task. We can see from Figure 6 B2 that the first layer filters are more primitive center-surround vs oriented Gabors like in AlexNet (Figure 6 A2). Something else to consider is that there are residual connections for ResNets, and it is possible that the multi-resolution inputs from Foveated-Blur implicitly sway the first layer filters to have a more low-resolution bias rather than a high-resolution bias. This difference is verified when comparing Low Pass Spatial frequency bias plots of 6 A1 vs 6 A2 (distance of green vs pink curves).
>
> * Clarity: [... Improve writing style for target audience (Computer Vision) and motivation for texture-based foveation transform]
>
> We agree and we will do our best with the extra page to do so. We hoped Figure 1 could introduce this theme gently, and we will try to find a way (perhaps via supplementary figures) to make it more accessible.
>
> * Significance: "do I know things that I didn't before? yes. Are they useful? that's completely unclear to me."
>
> Just to clarify: our goal was **not to create a model of human vision**. The goal of this paper is to **test** and **understand** if a foveated texture-based inductive bias (inspired by human peripheral vision) on a machine could give rise to different perceptual signatures than a set of 2 non-foveated inductive biases with different rates of compression, or a matched-resource adaptive-gaussian based inductive bias. Our goal was not to beat a state-of-the-art model as done in classical computer vision papers -- and neither to present a biologically plausible model of the human visual system to fit our model to behaviour as done in vision science; rather our goal is to show precise and empirical understanding of what are the potential consequences of different types of human-inspired visual processing (foveation) in machines -- and we think NeurIPS is the most appropriate audience for such type of works.
>
> We hope this has clarified many of the doubts presented in your review, and please let us know if something is not clear to potentially increase your score, as we are aiming our work to be accessible to multiple crowds. We think this is a paper that both provides strong experimental results and theoretical understanding of the mechanisms of foveation applied to scene classification that would be of great benefit to the Computer Vision, ML + Neuroscience communities at NeurIPS.

---

> > ### Comment · Reviewer_ThVH · 2021-08-27
> > **Acknowledging rebuttal**
> >
> > Thank you for the response; clearly you clarified some of the points raised originally, however to quote reviewer 89Wr "Overall, I am torn on this one"; you've done many experiments to investigate (well) the effect of a particular type of foveation, but it's still not clear to me what the point is. More specifically I think it seems to all reviewers that the choice of foveation model is so unusual (in the sense that its far from anything biologically plausible (or actually that close to human vision), or anything that would seem to be useful for machine vision) that we're having difficulty in understanding the real value of the results.

---

> > > ### Author Response · Authors · 2021-08-29
> > > **Follow-up Comments for Reviewer ThVH and AC**
> > >
> > >
> > > Dear AC and Reviewer ThVH,
> > >
> > > We think the informal consensus across reviewers that “the choice of foveation model [adaptive-texture] is so unusual [...]” proves one of the main motivations for our paper to be submitted to NeurIPS: **to raise awareness that this adaptive-texture based foveation is well known in the vision science literature, but has largely been ignored in computer vision and the ML community for the past decade(!)** as seen in the list of the following seminal works cited in our submission:
> > >
> > > * (1) A summary-statistic representation in peripheral vision explains visual crowding. Balas, Nakano & Rosenholtz. Journal of Vision, 2009.
> > >
> > > * (2) Metamers of the ventral stream. Freeman & Simoncelli. Nature Neuroscience, 2011.
> > >
> > > * (3) Visual Crowding. Pelli. Current Biology, 2011.
> > >
> > > * (4) What your visual system sees where you are not looking. Rosenholtz. Proceedings of SPIE (Human Vision and Electronic Imaging XVI), 2011.
> > >
> > > * (5) Rethinking the role of top-down attention in vision: Effects attributable to a lossy representation in peripheral vision. Rosenholtz, Huan & Ehinger. Frontiers in Psychology, 2012.
> > >
> > > * (6) Testing models of peripheral encoding using metamerism in an oddity paradigm. Wallis, Bethge & Wichmann. Journal of Vision, 2016.
> > >
> > > * (7) A general account of peripheral encoding also predicts scene perception performance. Ehinger & Rosenholtz. Journal of Vision, 2016.
> > >
> > > * (8) Capabilities and Limitations of Peripheral Vision. Rosenholtz. Annual Review of Vision Science, 2016.
> > >
> > > * (9) Selectivity and tolerance for visual texture in macaque V2. Ziemba, Freeman, Movshon & Simoncelli. PNAS, 2016.
> > >
> > > * (10) Mid-level visual features underlie the high-level categorical organization of the ventral stream. Long, Yu & Konkle. PNAS, 2018.
> > >
> > > * (11) Towards Metamerism via Foveated Style Transfer. Deza, Jonnalagadda & Eckstein. ICLR, 2019.
> > >
> > > * (12) Image content is more important than Bouma’s Law for scene metamers. Wallis, Funke, Gatys, Wichmann & Bethge. eLife, 2019.
> > >
> > > * (13) Opposing effects of selectivity and invariance in peripheral vision. Ziemba & Simoncelli. Nature Communications, 2021.
> > >
> > > And recently starting to make way in **computer graphics**:
> > >
> > > * (14) Beyond Blur: Real-time Ventral Metamers for Foveated Rendering. Watron, Kuffner dos Anjos, Friston, Swapp, Aksit, Steed & Ritschel. ACM Transactions on Graphics, 2021.
> > >
> > > * (15) Efficient Dataflow Modeling of Peripheral Encoding in the Human Visual System. Brown, DuTell, Walter, Rosenholtz, Shirley, Morgan & Luebke. ACM Transactions on Graphics, 2021.
> > >
> > > Indeed, **we think that the reviewers uncertainty about the biological plausibility of the foveation model could also be related to the variety of possible linking hypotheses between these kinds of computational models and the biological visual system**. To a systems neuroscientist, where this model stands as a model of neural mechanism, the biological plausibility is more relevant. To a **cognitive computational neuroscientist**, the models stand in different relation to the visual system, as a more *abstract perceptual architecture*, where the linking constructs are less about neurons and connections, and more about the nature of the learned representation, and its functional capacities and limits,  given different input and architectural constraints.
> > >
> > > **Critically for NeurIPS**: what is the value of this work to a vision scientist or machine vision researcher?
> > >
> > > To the **vision scientist**, these models are interesting and valuable because they provide clear tests of what kinds of functional consequences downstream visual representation might have when learning over a texture-based periphery.  This has long been a subject of debate and interest in the community (see references above), however the functional consequences of texturized periphery remained largely a subject for speculation; these models provide real opportunities to make concrete any consequences of a texturized periphery on higher order representation.  Indeed, even this preprint is being used in a Trends in Cognitive Science (TICS) paper, as the center-to-periphery organization of the visual system is becoming an increasingly important factor in understanding large-scale brain organization.  The point here is that for the visual cognitive neuroscience, **the objective is not to replicate the biology -- but to capture the hypothesized nature of the texturized periphery in order to turn just-so speculation into concrete understandings of possible functional consequences.**
> > >
> > > To the **machine vision researcher**, the models are interesting and valuable in that they reflect paths that are unlikely to be explored, were it not for the interdisciplinary influence of understanding human/primate vision. After all, with only a machine vision perspective (rather than a more interdisciplinary/NeurIPS perspective in mind), one might naively think, “why should we ever build a model that transforms/distorts information from the input?” and then never explore these questions  -- recall: Geirhos et al.'s Stylized Image-Net as a solution to the Texture-Bias which has developed onto it's own sub-field!  In a similar fashion, we take this question on, and **begin to chart the functional consequences of this texturized periphery, and more generally of different spatially-adaptive architectural constraints and computations in CNNs**. We reveal several insights, e.g. that this does not dramatically impair scene categorization ability, and that the texturized periphery induces a central-bias despite a weight sharing constraint -- this was not expected initially, though in retrospect is a reasonable outcome and thus has important implications for the understanding the interplay between central and peripheral processing of visual information.
> > >
> > > We hope these comments have clarified our position about the value of these results, our choices in how we operationalized this question with our 2-stage models, and the biological plausibility of the adaptive-texture based foveation transform. Thank you once again for your time, perspective, and your consideration in reviewing our paper!

---

### Official Review · Reviewer_qxyi · 2021-07-19

**Rating:** 8
**Confidence:** 5

**Summary:**

The authors address hypotheses about the nature of peripheral biological vision, specifically the Foveation-Texture theory stating that peripheral representations in primate vision are not simply blurred (subsampled) but rather extract more informative texture statistics. The empirical study comparing the proposed model vs several baseline model corroborates their proposal.


**Limitations And Societal Impact:**

The authors of the study seems to have made a peculiar choice that might be undermining their conclusions: the first (texture processing) system that imitates foveation (and texture-level processing) actually acts as an encoder-decoder that generates processed images for the second stage, implemented as a standard CNN. However, a  more natural for both ML and for modeling a biological vision system would be to actually use the learnt latent codes of the autoencoder as the input to the CNN. So it would be interested to see how (if at all) the reaults would be affected by such processing pipeline.

Impact: better understanding of human vision with implications for medicine and novel inductive biases for computer vision models.

**Main Review:**

This work addresses an important question: Is foveated vision an artifact of constrained resources or does it endow primate vision with an advantage in terms of learning more efficient representations?
Their empirical studies corroborate the notion that texture-foveation mechanism offers greater generalization, preserves more high spatial frequency information in periphery and develops greater robustness to occlusion than the blur-based model of peripheral representations.
The paper is well written and the claims are convincingly supported by empirical tests.
However, the line of argumentation seems a bit confounded: while the authors state in the introduction that they focus on representational consequences of foveation, they are less interested in general such consequences of any foveation model as compared to non-foveated alternatives, but rather differences between specifically texture-foveation model vs other foveation models. It would be interesting to also address a model where the texture representation is used without the foveation model for testing these factors independently.

**Time Spent Reviewing:**

6

---

> ### Author Response · Authors · 2021-08-10
> **Response to Reviewer qxyi [1/1]**
>
>
> Thank you for your enthusiastic review \& perspective! We'd like to address one small question with regards to texture-based controls:
>
> * "It would be interesting to also address a model where the texture representation is used without the foveation model for testing these factors independently."
>
> This is a great idea, and we have done some preliminary experiments on this condition (and are polishing them for a follow-up paper)! In such preliminary experiments we found that the texture-trained network did not achieve performance anywhere near the 4 main systems in the i.i.d regime (did about 35%). In addition all 4 systems could marginally decode scene category information from texture-based scenes above chance at 20\% accuracy on Gatys Textures [chance is 5%]. We are also currently training and testing these networks on Gatys et al. based textures (Neural Synthesis), Huang \& Belongie textures (AdaIN) and Portilla \& Simoncelli textures.
>
> We are also currently working on a follow-up paper where we will have additional controls such as a Vision Transformer (ViT) as $g(\circ)$, robustness to PGD-based adversarial attacks $(L_\infty,L_2, L_1)$ and to test for a shape-vs-texture bias in scene perception.

---

### Decision · Program_Chairs · 2021-09-27

**Decision:**

Reject

**Comment:**

This paper received one positive review and two negative reviews (including one borderline reject). The positive reviewer did not engage in follow-up discussion which is based only on the two negative reviewers. There seems to have been some difficulty in communicating between the reviewers and the authors. As they stated in their final comments, I think the authors are right "perhaps there is a confusion on the goal (rather than the method) of our paper?". I agree that the reviewers understand what the methods is. However, it is pretty clear that they do not see the benefit of doing so either from an ML or neuroscience perspective. Even the positive reviewer calls the validity of the approach into question. Unfortunately it seems that no amount of revision will address this issue. For all these reasons, the AC recommends this paper being rejected.